# Broca's cerebral asymmetry reflects gestural communication's lateralisation in monkeys (*Papio anubis*)

Yannick Becker[1,2], Nicolas Claidière[1], Konstantina Margiotoudi[1], Damien Marie[1], Muriel Roth[2], Bruno Nazarian[2], Jean-Luc Anton[2], Olivier Coulon[3], Adrien Meguerditchian[1,4]*

[1]Laboratoire de Psychologie Cognitive, CNRS, Aix-Marseille University, Marseille, France; [2]Centre IRMf Institut de Neurosciences de la Timone CNRS, Aix-Marseille University, Marseille, France; [3]Institut de Neurosciences de la Timone CNRS, Aix-Marseille University, Marseille, France; [4]Station de Primatologie CNRS-CELPHEDIA, Marseille, France

**Abstract** Manual gestures and speech recruit a common neural network, involving Broca's area in the left hemisphere. Such speech-gesture integration gave rise to theories on the critical role of manual gesturing in the origin of language. Within this evolutionary framework, research on gestural communication in our closer primate relatives has received renewed attention for investigating its potential language-like features. Here, using in vivo anatomical MRI in 50 baboons, we found that communicative gesturing is related to Broca homologue's marker in monkeys, namely the ventral portion of the Inferior Arcuate sulcus (*IA sulcus*). In fact, both direction and degree of gestural communication's handedness – but not handedness for object manipulation are associated and correlated with contralateral depth asymmetry at this exact *IA sulcus* portion. In other words, baboons that prefer to communicate with their right hand have a deeper left-than-right *IA sulcus*, than those preferring to communicate with their left hand and vice versa. Interestingly, in contrast to handedness for object manipulation, gestural communication's lateralisation is not associated to the *Central sulcus* depth asymmetry, suggesting a double dissociation of handedness' types between manipulative action and gestural communication. It is thus not excluded that this specific gestural lateralisation signature within the baboons' frontal cortex might reflect a phylogenetical continuity with language-related Broca lateralisation in humans.

*For correspondence:
adrien.meguerditchian@univ-amu.fr

**Competing interest:** The authors declare that no competing interests exist.

## Editor's evaluation

This is an elegant, well-designed study, filling a gap regarding structural asymmetries in Broca's area for Old World monkeys. Using a good number of subjects (50 baboons), the authors build on earlier behavioral work pointing to handedness contrasts between communicative vs manipulative gestures, and tested whether the lateralisation effect associated specifically with communicative gestures manifested itself in the depth asymmetry of the ventral Inferior Arcuate [IA] sulcus [Broca's area homologue], but not in the central sulcus. The results of the experiments indeed show a dissociation between communicative vs manipulative gestures, with only the communicative gestures being associated with an IA sulcus depth asymmetry tracking lateralized hand use. The authors point to a captivating phylogenetic continuity between language lateralization in humans and brain asymmetries related to gestures in nonhuman primates.

## Introduction

Broca's area and its left hemispheric specialisation has historically been considered as the centre of speech production. Even if such a modular conception of language's neural bases was questioned by models of plastic and large distributed networks (*Friederici, 2017*; *Hickok and Poeppel, 2007*), it is still well acknowledged that Broca's area remains a key node for language specialisation within this neural distributed network. Complementary work thereby highlighted Broca's area as an interface between speech and multimodal motor integration including gesture and mouth mouvements (*Gentilucci and Dalla Volta, 2008*). Broca's area is also known for its involvement in motor planning, sequential, and hierarchical organization of behaviours, such as linguistic grammar or tool use and tool making (*Gentilucci and Dalla Volta, 2008*; *Koechlin and Jubault, 2006*; *Stout et al., 2015*; *Corballis, 2015*; *Wakita, 2014*). This body of work raises evolutionary questions about the role of the motor system and gestural communication in language origins and its brain specialisation. Therefore, a growing number of researchers proposed that language organisation took some of its phylogenetical roots into a gestural system across primate evolution (*Gentilucci and Dalla Volta, 2008*; *Corballis, 2015*; *Tomasello, 2008*). Consequently, whereas comparative language research has focused on the potential continuities across primate brain circuitry (e.g., *Balezeau et al., 2020*; *Becker et al., 2022*) or vocal and auditory systems (e.g. *Boë et al., 2017*; *Jarvis, 2019*; *Wilson et al., 2017*), the research on gestural communication in apes and monkeys has historically shown a significant interest within this evolutionary framework.

A large body of nonhuman primate studies has documented some continuities of the communicative gestural system with several key features of human language such as intentionality, referentiality, learning flexibility, and lateralisation (e.g. *Tomasello, 2008*; *Meguerditchian and Vauclair, 2014*; *Molesti et al., 2020*). About manual lateralisation specifically, studies in baboons and great apes have indeed shown that communicative manual gesturing elicited stronger right-hand use in comparison to non-communicative manipulative actions at a population-level (reviewed in: *Meguerditchian et al., 2013*). In addition, at the individual level, a double dissociation concerning the type of handedness has been documented between gestural communication and object manipulation, showing that primates classified as right-handed for communicative gesture are not especially classified as right-handed for object manipulation and vice versa (*Meguerditchian and Vauclair, 2006*). Those behavioural findings in different primate species suggested a specific lateralised system for communicative gestures, which might be different from the one involves in handedness for object manipulation. This is consistent with human literature showing that typical object-manipulation handedness measures turned out to be a rather poor marker of language lateralisation (*Fagard, 2013*), as most left-handers (78%) also show left-hemisphere dominance for language (*Knecht et al., 2000*; *Mazoyer et al., 2014*), just like right-handed people. In both humans and nonhuman primates, direction of handedness for object manipulation was found associated to contralateral asymmetries of the motor hand area within the *Central sulcus* (e.g. humans: *Amunts et al., 2000*; *Cykowski et al., 2008*; chimpanzees: *Hopkins and Cantalupo, 2004*; *Dadda et al., 2006*; Baboons: *Margiotoudi et al., 2019*; Capuchin monkeys: *Phillips and Sherwood, 2005*; Squirrel monkeys: *Nudo et al., 1992*). In fact, it has recently been demonstrated that the neural substrates of typical handedness measures and language brain organisation might be not related but rather independent from each other (*Groen et al., 2013*; *Ocklenburg et al., 2014*; *Häberling et al., 2016*; *Labache et al., 2020*).

Whether gestural communication's handedness in humans is a better predictor of language lateralisation and is thus different than typical handedness measures remain unclear. Nevertheless, several studies in humans are supporting this hypothesis. In early human development, the degree of right-handedness for preverbal gestures is more pronounced at a population-level than handedness for manipulation (*Blake et al., 1994*; *Bonvillian et al., 1997*; see also *Fagard, 2013*; *Cochet and Vauclair, 2010*) and increases when the lexical spurt occurs in children contrary to manipulation handedness (*Cochet et al., 2011*). Moreover, further work showed Broca's activation in the left hemisphere also for sign language production including manual and oro-facial gestures (*Emmorey et al., 2004*; *MacSweeney et al., 2008*).

Given such potential lateralisation links between gesture and language in humans, it is thus not excluded that the specific lateralisation's signature found for communicative gestures in nonhuman primates might reflect evolutionary continuities with frontal hemispheric specialisation for speech/gesture integration. This hypothesis might be relevant to investigate given brain studies in nonhuman

primates have shown gross left-hemispheric asymmetries of homologous language areas at a populational level that are similar to the ones described in humans (e.g., *Geschwind and Levitsky, 1968*; *Keller et al., 2009*): In particular Broca's homologue in great apes (*Cantalupo and Hopkins, 2001*; *Graïc et al., 2020*) as well as the Planum Temporale in great apes and even in baboons, an Old World monkey species, in both adult and newborns (*Gannon et al., 1998*; *Marie et al., 2018*; *Becker et al., 2021a*; *Becker et al., 2021b*).

For Old World monkeys specifically, no study regarding structural asymmetry for Broca's homologue has been investigated. One reason is that determining this area in monkeys is particularly challenging in comparison to apes. In fact, the inferior precentral sulcus, the inferior frontal sulcus and the fronto-orbital sulcus, which are common borders of Broca's homologue in apes (*Cantalupo and Hopkins, 2001*), are absent in monkeys and thus delimitation is not trivial. Nevertheless, all the detailed cytoarchitectonic studies addressing the Broca's homologue within the frontal lobe in Old World monkeys (i.e. in mostly macaques but also in baboons) – and its two components Area 44 and 45 – pointed towards the same sulcus of interest as the epicentre of this region: the mid-ventral and ventral portion of the Inferior Arcuate sulcus (*IA sulcus*). The *IA sulcus* is considered homologue to the ascending branch of the inferior precentral sulcus (*Amiez and Petrides, 2009*) which delimits Broca's area posteriorly in humans and great apes. In monkeys, Area 45 homologue sits in the anterior bank of the ventral *IA sulcus* (*Petrides et al., 2005*). In contrast, Area 44 homologue might be located in the fundus and the posterior bank of the ventral *IA sulcus* in monkeys (*Petrides et al., 2005*), which overlaps with F5 region related to the mirror neuron system (*Belmalih et al., 2009*; *Rizzolatti and Fogassi, 2014*). Electric stimulation in the fundus of the ventral *IA sulcus* elicited oro-facial and finger mouvements in macaques (*Petrides et al., 2005*). Concerning baboons specifically, a cytoarchitectonic study (*Watanabe-Sawaguchi et al., 1991*) showed similarities to the macaque frontal lobe organisation given Area 45 anteriorly to the *IA sulcus*, even if Area 44 was not described (*Petrides et al., 2005*; *Belmalih et al., 2009*; *Rizzolatti and Fogassi, 2014*; *Watanabe-Sawaguchi et al., 1991*). Therefore, in the absence of the usual Broca's sulcal borders found in apes, the depth of the ventral part of the *IA sulcus* constitutes the only critical neuroanatomical marker for delimiting the border and the surface of Broca's homologue in monkeys.

In sum, within the framework of the origin of hemispheric specialisation for language, most comparative works in nonhuman primates focused on population-level asymmetry for either brain or communicative behaviours. Those studies have reported similar population-level leftward brain asymmetry for key language homologue regions (*Gannon et al., 1998*; *Cantalupo and Hopkins, 2001*; *Marie et al., 2018*; *Becker et al., 2021a*; *Becker et al., 2021b*) or similar populational-level right-handedness for communicative gestures (reviewed in: *Meguerditchian et al., 2013*). Nevertheless, to test potential phylogenetic continuities, this approach suffered from lack of studies investigating the direct behavioural/brain correlates at the individual-levels.

In the present in-vivo MRI study conducted in 50 baboons (*Papio anubis*), we have (1) measured the inter-hemispheric asymmetries of the *IA sulcus*' depth – from its dorsal to its most ventral portion among subjects for which the *Central sulcus* depth measure was available from a previous study (*Margiotoudi et al., 2019*) (2) as well as its potential links with direction and degree of communicative gesture's handedness in comparison to handedness for manipulative actions as measured with a bimanual tube task (see *Hopkins, 1995*). In other words, we tested specifically whether the depth asymmetry of the most ventral Inferior Arcuate sulcus' portion (ventral *IA sulcus*, i.e. the Broca's homologue) – but not the *Central sulcus* – was exclusively associated with communicative gestures' lateralisation.

## Results

Between baboons communicating preferentially with the right hand *versus* the ones with the left hand, we found significant contralateral differences of depth asymmetries in the ventral portion of the *IA sulcus* (i.e., from the mid-ventral IA position to the most ventral *IA sulcus* portion, namely from contiguous positions 65–95 out of the 99 segmented positions of the entire *IA sulcus*) according to a cluster-based permutation test ($p < .01$, t-value clustermass = 76.16, for $p < .01$ a clustermass of 65.28 was needed, see *Maris and Oostenveld, 2007*). In other words, the 28 baboons using preferentially their right hand for communicative gestures showed more leftward *IA sulcus* depth asymmetry at this cluster than the 22 ones using preferentially their left-hand. In contrast, for non-communicative

manipulative actions, we found no significant difference of sulcus depth asymmetries between the left- (N = 22) *versus* right-handed (N = 28) groups concerning any portion of the *IA sulcus*, according to a cluster-based permutation test ($p > .10$) (*Figure 1*).

In addition, after calculating the AQ score per subject representing the sulcus depth asymmetry of the whole 'Broca's cluster' (i.e. from the sum of the *IA sulcus* depths from positions 65–95 in the left hemisphere and the sum of *IA sulcus* depths from position 65–95 in the right hemisphere), we found a significant negative correlation between individual AQ depth values of the Broca's cluster (i.e. from positions 65–95) and individual handedness degree for communication (HI): $r(48) = −.337$; $p < .05$ (i.e. The stronger the hand preference is for one hand, the deeper is the *IA sulcus* asymmetry from positions 65 to 95 in the contralateral hemisphere) (*Figure 2A*). In contrast, AQ depth values of the Broca's cluster did not show significant correlation with HI for non-communicative actions ($r(48) = −.037$; $p ≈ 1$) (*Figure 2B*). Using the cocor package in R (*Diedenhofen and Musch, 2015*), a comparison between these two overlapping correlations based on dependent groups showed a significant difference between the two correlations ($p < .05$).

When comparing with the control sulcus of interest, the *Central sulcus* related to the primary motor cortex, an opposite effect was found between handedness for manipulative actions and hand preferences for communicative gesture. We found no significant difference of sulcus depth asymmetries regarding communicative gestures. In contrast, *Margiotoudi et al., 2019* reported that the CS presented a contralateral asymmetry at continuous positions 56–60 (labelled as the 'Motor-hand area's cluster') for non-communicative manipulative actions, after permutation tests for correction (*Figure 1*).

Finally, we conducted a mixed-model analysis of variance with AQs depth values for the *IA sulcus* 'Broca's cluster' and for the *Central sulcus* 'Motor hand area's cluster' (AQ derived from continuous positions 56–60, see *Margiotoudi et al., 2019*) serving as the repeated measure while communication handedness (left- *versus* right-handed) and action handedness (left- *versus* right-handed) were between-group factors. The mixed-model analysis of variance demonstrated a significant main effect on the AQ scores for communication handedness ($F1_{,46} = 14.08$, $p < .01$) and for action handedness ($F_{1,46} = 4,1$, $p < .05$).

## Discussion

The results of the study are straightforward. We showed that the *IA sulcus* left- or rightward depth asymmetry at its mid-ventral and ventral portion (labelled as the 'Broca cluster') is associated exclusively with contralateral direction (left-/right-hand) of communicative manual gestures' lateralisation in baboons but not handedness for non-communicative actions. Building upon these first results, we also found a significant negative correlation between the Handedness Index (HI) values for communicative gestures and the Asymmetric Quotient (AQ) depth values of the *IA sulcus* 'Broca's cluster', suggesting that the contralateral links between handedness for gestural communication and depth asymmetries at the most ventral portion of the *IA sulcus* is evident not only at a qualitative level but also at a quantitative level as well. In other words, individuals with a stronger degree of manual lateralisation for communicative gesture have greater *IA sulcus* depth asymmetries at this ventral cluster in the hemisphere contralateral to their preferred hand for communication. The ventral positions of such sulcal depth asymmetries are clearly at a crossroad of Broca-related frontal regions including the fundus of the sulcus, Area 44 (*Petrides et al., 2005*), the anterior bank, Area 45 (*Petrides et al., 2005*), the posterior bank and ventral F5 or granual frontal area (GrF) (*Belmalih et al., 2009*; *Rizzolatti and Fogassi, 2014*). Since the sulcus' depth might reflect a gyral surface and its underlying grey matter volume, future work of delineating and quantifying grey matter of the ventral *IA sulcus* would help determining which of those sub-regions of the Broca homologue is driving the asymmetry specifically, for instance by VBM methods.

Whereas handedness for manipulative actions in baboons was previously found related to the motor cortex asymmetry within the *Central sulcus* (*Margiotoudi et al., 2019*), our present findings report the first evidence in monkeys that the neurostructural lateralisation's landmark of communicative gesture is located in a frontal region, related to Broca's homologue. Such a contrast of results between manipulation and communication found at the cortical level is consistent with what was found at the behavioural level in studies showing that communicative gesture in baboons and chimpanzees elicited specific and independent patterns of manual lateralisation in comparison to non-communicative

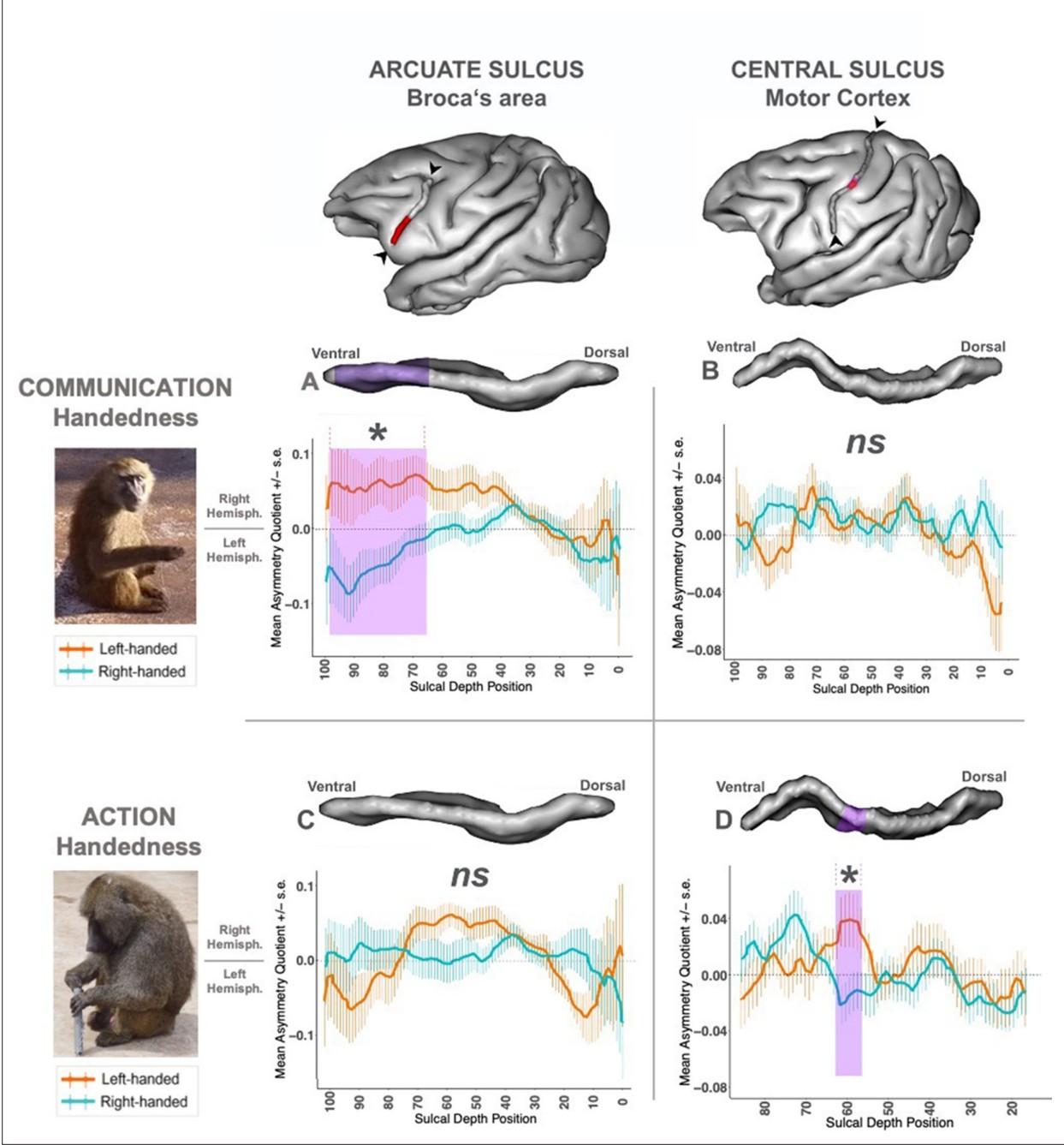

**Figure 1.** Effect of left-/right-hand direction of two handedness types (communication vs action) on neuroanatomical sulcus depth asymmetries (*IA sulcus* vs *Central sulcus*). *Left panel*: Pictures of the two types of handedness measures in baboons. 'Communication Handedness': a 'Handslap' communicative gesture in a juvenile male; 'Action Handedness': the non-communicative bimanual coordinated 'tube task' performed by an adult male. *Top panel*: 3-D brain representation from BrainVisa software of the baboon's left hemisphere, including the *IA sulcus*; and the *Central sulcus* with the portion in purple where a significant effect was found in ***Margiotoudi et al., 2019***. *Graphs*: Sulcus depth's asymmetry (AQ) comparison between right-handed group *versus* left-handed group of baboons classified according to the type of manual tasks. Positive Mean Asymmetry Quotient values (AQ) indicate rightward hemispheric asymmetry, negative Mean Asymmetry Quotient values leftward hemispheric asymmetry. +/- SE indicated the Standard Error. (**A**) *IA sulcus* AQ between right-handed (N=28) *versus* left-handed (N=22) groups' classification for communicative 'Handslap' gesture. Significant contralateral AQ difference (*p* < .01) between the two groups was found for a cluster including positions 65 to 95 (highlighted in purple in the graph and the 3D representation of the *IA Sulcus*). (**B**) *Central Sulcus* AQ between right-handed (N=28) *versus* left-handed (N=22) groups' classification for non-communicative bimanual coordinated actions. (**C**) *IA sulcus* AQ between right-handed (N=28) *versus* left-handed (N=22) groups' classification for non-communicative bimanual coordinated actions. (**D**) Initial graph (Adapted from Figure 2 from ***Margiotoudi et al., 2019***) of the *Central Sulcus* AQ showing the significant contralateral AQ differences (*p* < .05) found between the left-handed (N=28) *versus* right-handed (N=35) groups group for the

*Figure 1 continued on next page*

*Figure 1 continued*

non-communicative bimanual coordinated actions (Action condition) for positions 56 to 61 (highlighted in purple in the graph and the 3D representation of the *Central Sulcus*).

manipulative actions (*Meguerditchian and Vauclair, 2009*; *Meguerditchian et al., 2010*). Therefore, it provides additional support to the hypothesis suggesting that gestural communication's lateralisation in nonhuman primates might be, just as language brain organisation in human (see *Häberling et al., 2016*), related to a different lateralised neural system than handedness for pure manipulative action. Its specific correlates with Broca's homologue's lateralisation is also consistent with what was found in our closest relatives, the chimpanzee (*Taglialatela et al., 2006*; *Meguerditchian et al., 2012*).

Regarding Broca's area in humans, very recently, a functional segregation was proposed with Broca's anterior part implicated in language syntax and its posterior part exclusively implicated in motor actions (*Zaccarella et al., 2021*). The authors argued that action and language meet at this interface. In an evolutionary perspective we propose therefore that the intentionality of primate's communicative gesture might account for this hypothesised functional interface of actions and language prerequisites, nested inside the monkeys' Broca's homologue (see also: *Rizzolatti and Craighero, 2004Arbib et al., 2008*; *Rizzolatti and Fogassi, 2014*; *Corballis, 2015*). In addition, in macaques Broca's homologue, neuronal recordings showed populations of specific neurons activated for both volitional vocal and manual actions (*Gavrilov and Nieder, 2021*).

The articulation of our results with this recent literature suggests that gestural communication may be a compelling modality for one of the multimodal evolutionary roots of the typical multimodal language system in humans and its hemispheric specialisation. It is thus not excluded that language-related frontal lateralisation might be much older than previously thought and inherited from a gestural communicative system dating back, not to Hominid origins, but rather to the common ancestor of humans, great apes and Old World monkeys, 25–35 million years ago.

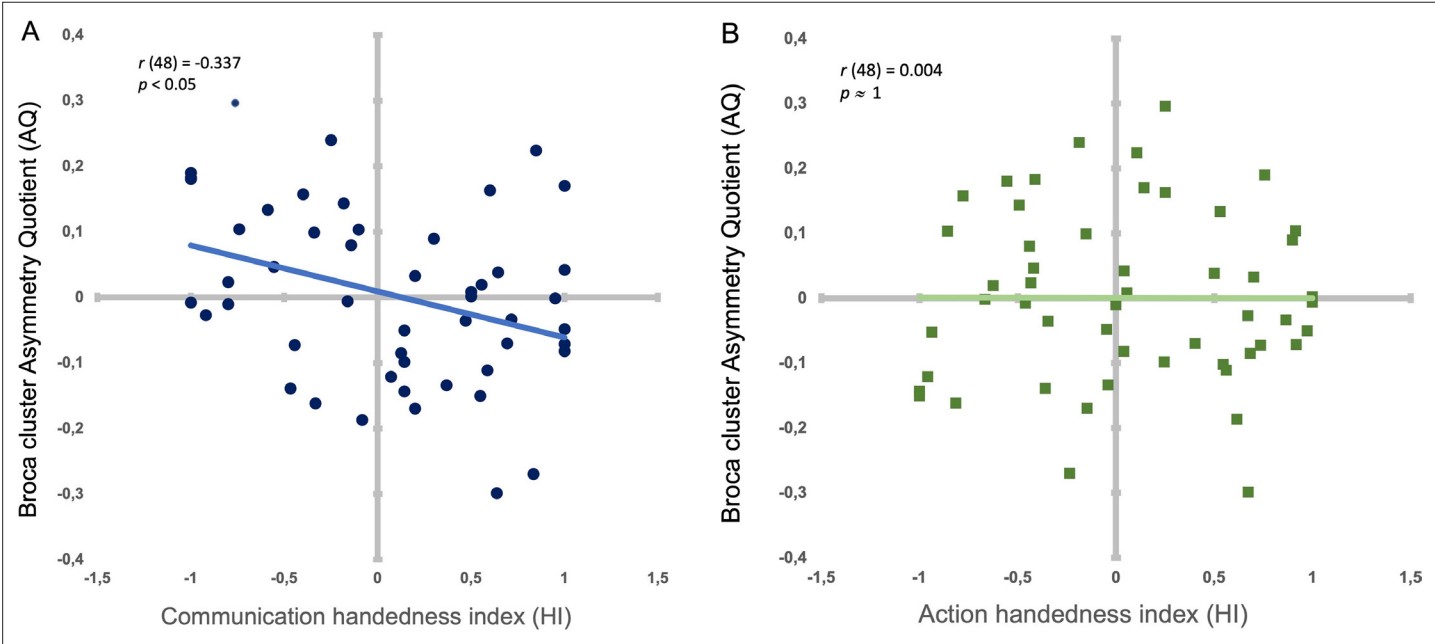

**Figure 2.** Correlation between handedness degree types and the Broca's cluster's asymmetry. (**A**) Individual handedness degree (HI) for communicative gestures and AQ depth values of the Broca's cluster (i.e. from positions 65–95) in dark blue dots. Light blue line: Significant negative correlation between HI and AQ. B. Individual handedness degree (HI) for non-communicative manipulative actions (HI) and AQ depth values of the Broca's cluster (i.e. from positions 65–95) in dark green squares. Light green line (superposing on x axis): Non-significant correlation between HI and AQ.

# Materials and methods

## Subjects

Inter-hemispheric asymmetries of the *IA sulcus*' depth were quantified from anatomical T1w MRI images in 80 baboons *Papio anubis* born in captivity and free from developmental or anatomical abnormalities or brain disorders (generation F1, 52 females, 28 males, age range = 7–32 years, mean age (years): $M$ = 17.7, SE = 5.9). Out of this sample, were included only subjects which overlaps with both (1) the sample of subjects for which individual measures of handedness for communicative gestures were available (i.e. hand slapping gesture, *Meguerditchian and Vauclair, 2006*) and (2) the previous sample of 63 subjects (i.e. 35 right-handed and 28 left-handed) reported in *Margiotoudi et al., 2019* for which both *Central sulcus* depth measures and individual measures of handedness for manipulative actions (i.e. the bimanual tube task, *Vauclair et al., 2005*) were reported. It resulted a total overlap of 50 baboons (29 females and 21 males, mean age (years): $M$ = 12.3, SE = 5.8) who combined thus the both types of measures of handedness (communication *versus* manipulation) and the depth measures of the two sulci of interest (*IA sulcus* and *Central sulcus*) in the two hemispheres of the brain.

All baboons were housed in social groups at the Station de Primatologie CNRS (UPS 846, Rousset, France; Agreement number for conducting experiments on vertebrate animals: D13-087-7) and have free access to outdoor areas connected to indoor areas. Wooden and metallic, ethologically approved, structures enrich the enclosures. Feeding times are held four times a day with seeds, monkey pellets and fresh fruits and vegetables. Water is available ad libitum. The study was approved by the 'C2EA-71 Ethical Committee of Neurosciences' (INT Marseille) under the number APAFIS#13553–201802151547729. The experimental procedure complied with the current French laws and the European directive 86/609/CEE.

## Sulcal parametrization

The *IA sulcus* and the *Central sulcus* were extracted from T1w images using the pipeline of the free BrainVisa software (see *Mangin et al., 2004* for details of the procedure). The sulcus parametrization tool within the BrainVisa toolbox provides therefore sulcus-based morphometry by subdividing the *sulci* of each hemisphere into 99 standardized positions from dorsal to ventral sulcus extremities in order to quantify the variation of sulcal depth all across the *sulci's* 99 positions (*Mangin et al., 2004*). This automatic algorithm is free from observer's judgment. To estimate asymmetries of the *sulci's* depth between the two hemispheres an asymmetry quotient (AQ) for each of the 99 sulcal positions AQ = (R – L) / [(R + L × 0.5)] was computed (*Margiotoudi et al., 2019*). The AQ values vary between –2 and +2 with positive values indicating right-hemispheric lateralisation and negative values indicating left-hemispheric lateralisation.

## Behaviour correlate

For further investigating its potential behavioural correlates, we tested whether the right- *versus* left-handed groups classified for a given manual task (i.e. gestural communication *versus* manipulative actions) differed in term of neurostructural depth asymmetries (AQ) within the *IA sulcus* and the *Central sulcus*. The two types of handedness measures were previously collected (for communicative gesture: *Meguerditchian and Vauclair, 2006*; *Meguerditchian et al., 2011* for manipulative actions: *Vauclair et al., 2005*; *Molesti et al., 2016*).

Communicative gesture was defined as a mouvement of the hand directed to a specific partner or audience in order to affect its behaviour (*Molesti et al., 2020*). Like in apes, some communicative manual gestures in baboons have been found to share human-like intentional control, referential properties, flexibility of acquisition and of use as well as similar specific pattern of manual lateralisation (reviewed in *Tomasello, 2008*; *Meguerditchian and Vauclair, 2014*; *Meguerditchian et al., 2013*). The present study focused specifically on the 'Hand slapping' gesture which was previously found optimal for measuring gestural communication's lateralisation in this species (*Meguerditchian and Vauclair, 2006*; *Meguerditchian et al., 2011*). Indeed, the hand slapping behaviour – a probably innate gestures used by the baboon to threat or intimidate the recipient – is the most common and frequent visual gesture of the repertoire (*Molesti et al., 2020*) produced intentionally and unimanually in a lateralised manner across similar agonistic contexts and similar emitter's postures (*Meguerditchian et al., 2013*). Hand use was recorded in a baboon when slapping or rubbing quickly and

repetitively the hand on the ground in direction to a conspecific or a human observer at an out of reach distance. Recorded events were taken from different bouts and not repeated measures from the same bout. As *Margiotoudi et al., 2019*, in case a subject has been assessed in multiple sessions within 2004–2015, the final classification as right or left-handed was selected based on the session with the most observations, excluding subjects with less than five observations (*Mean = 25.98, S.E. = 3.67*).

Handedness for manipulative actions was assessed using the well-documented bimanual coordinated 'Tube task' (*Hopkins, 1995*). Hand use was recorded when extracting food with a finger out of a PVC tube hold by the other hand.

The individual handedness index (HI) for a given manual behaviour, or degree of individual manual asymmetry, was calculated based on the formula (#R-#L)/(#R+#L), with #R indicating right hand responses and #L for left hand responses. The HI values vary between –1 and +1 with positive values indicating right hand preference, negative values indicating left hand preference and 0 indicating no preference. The absolute HI score indicate the strength of manual preference.

Among the 50 baboons, for communicative gesture, 22 subjects were thus classified as left-handed, 28 as right-handed following the HI direction. A 51[th] subject, having a HI score of 0 (i.e. no manual bias), could not be classified in either categories and has been thus excluded from the study. For object-related manipulative actions (i.e. the bimanual tube task), 22 subjects were classified as left-handed, 28 as right-handed as already reported in *Margiotoudi et al., 2019* for those 50 overlapping subjects. Among the 50 baboons, 18 subjects switched left-/right-handed categories of hand preference between communicative gesture and manipulative actions (i.e. 9 from left-handed group for gestural communication to right-handed group for manipulative actions, 9 from right-handed group for gestural communication to left-handed group for manipulative actions).

## Statistical analysis

Statistical analysis was conducted using R 3.6.1 by Cluster Mass Permutation tests (*Maris and Oostenveld, 2007*). First, an assembly of depth asymmetry measures was defined as a 'cluster' when continuous significant differences of the same sign across positions were found between groups (two-sided t-tests, Welch corrected for inequality of variance, $p < .05$). Second, the sum of t-values within each cluster was calculated (the 'cluster mass'). Next, permutations were conducted for the between individual tests: For a given type of manual behaviour, Left-handed individuals' AQ values *versus* Right-handed individuals' AQ values were randomly redistributed between individuals and the maximum absolute cluster mass was calculated for each randomly permuted set. This procedure was repeated 5000 times and the 99% confidence interval (CI) of the maximum cluster mass was calculated. The clusters in the observed data were considered significant at 1% level if their absolute cluster mass was above the 99% CI of the distribution (i.e. $p < .01$).

We also performed a linear correlation between (1) the Handedness Index (HI) values for communicative gesture calculated from the 50 individuals and (2) the Asymmetric Quotient (AQ) values of those 50 baboons calculated from the respective left and right ventral *IA sulcus'* depth sum of the continuous positions of the cluster for which a significant difference in AQ score is detected by t-test comparison between the right- and left-handed groups. The same procedure was followed for the HI values for non-communicative actions for those 50 individuals.

## Acknowledgements

We are very grateful to the Station de Primatologie CNRS, particularly the animal care staff and technicians, Jean-Noël Benoit, Jean-Christophe Marin, Valérie Moulin, Fidji and Richard Francioly, Laurence Boes, Célia Sarradin, Brigitte Rimbaud, Sebastien Guiol, Georges Di Grandi for their critical involvement in this project; Leonard Samain-Aupic for sulci labelling; Ivan Balansard, Sandrine Melot-Dusseau, Laura Desmis, Frederic Lombardo and Colette Pourpe for additional assistance.

# Additional information

## Funding

| Funder | Grant reference number | Author |
|---|---|---|
| H2020 European Research Council | 716931 - GESTIMAGE - ERC-2016-STG | Adrien Meguerditchian |
| Agence Nationale de la Recherche | ANR-12-PDOC-0014-01 | Adrien Meguerditchian |
| Agence Nationale de la Recherche | ANR-16-CONV-0002 | Adrien Meguerditchian |
| Initiative d'Excellence d'Aix-Marseille Université | A*Midex AMX-19-IET-004 | Yannick Becker |
| Agence Nationale de la Recherche | ANR-17-EURE-0029 | Yannick Becker |
| Agence Nationale de la Recherche | LangPrimate | Adrien Meguerditchian |
| Agence Nationale de la Recherche | ILCB | Adrien Meguerditchian |
| Initiative d'Excellence d'Aix-Marseille Université | NeuroMarseille | Adrien Meguerditchian |
| Agence Nationale de la Recherche | NeuroSchool | Yannick Becker |

The funders had no role in study design, data collection and interpretation, or the decision to submit the work for publication.

## Author contributions

Yannick Becker, Data curation, Formal analysis, Writing - original draft; Nicolas Claidière, Damien Marie, Data curation, Formal analysis; Konstantina Margiotoudi, Formal analysis; Muriel Roth, Resources, designed MRI sequences; Bruno Nazarian, Resources, designed the baboons' monitoring programs; Jean-Luc Anton, Resources, coordinated the MRI sessions; Olivier Coulon, Methodology, Resources, Software, Writing - review and editing, designed the sulcus parametrization tool; Adrien Meguerditchian, Conceptualization, Funding acquisition, Investigation, Methodology, Project administration, Supervision, Writing - original draft, Writing - review and editing, supervised the MRI acquisitions

## Author ORCIDs

Yannick Becker ⓘ http://orcid.org/0000-0002-9728-8316
Konstantina Margiotoudi ⓘ http://orcid.org/0000-0002-9505-3978
Adrien Meguerditchian ⓘ http://orcid.org/0000-0003-3754-6747

## Ethics

All baboons were housed in social groups at the Station de Primatologie CNRS (UPS 846, Rousset, France; Agreement number for conducting experiments on vertebrate animals: D13-087-7) and have free access to outdoor areas connected to indoor areas. Wooden and metallic, ethologically approved, structures enrich the enclosures. Feeding times are held four times a day with seeds, monkey pellets and fresh fruits and vegetables. Water is available ad libitum. The study was approved by the "C2EA-71 Ethical Committee of Neurosciences" (INT Marseille) under the number APAF-IS#13553-201802151547729. The experimental procedure complied with the current French laws and the European directive 86/609/CEE.

## Decision letter and Author response

Decision letter https://doi.org/10.7554/eLife.70521.sa1
Author response https://doi.org/10.7554/eLife.70521.sa2

# Additional files

## Supplementary files
• Transparent reporting form

## Data availability
The behavioural, neuro-anatomical and statistic code data that support the findings of this study are available in "OSF Storage" with the identifier https://doi.org/10.17605/OSF.IO/DPXS5 and the URL https://osf.io/dpxs5/?viewonly=f406ad972edd43e485e5e4076bae0f78.

The following dataset was generated:

| Author(s) | Year | Dataset title | Dataset URL | Database and Identifier |
|-----------|------|---------------|-------------|-------------------------|
| Becker Y | 2020 | OSF Storage | https://osf.io/dpxs5/?viewonly=f406ad972edd43e485e5e4076bae0f78 | Open Science Framework, 10.17605/OSF.IO/DPXS5 |

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
