## [Editor Report]

This is an elegant, well-designed study, filling a gap regarding structural asymmetries in Broca's area for Old World monkeys. Using a good number of subjects (50 baboons), the authors build on earlier behavioral work pointing to handedness contrasts between communicative vs manipulative gestures, and tested whether the lateralisation effect associated specifically with communicative gestures manifested itself in the depth asymmetry of the ventral Inferior Arcuate [IA] sulcus [Broca's area homologue], but not in the central sulcus. The results of the experiments indeed show a dissociation between communicative vs manipulative gestures, with only the communicative gestures being associated with an IA sulcus depth asymmetry tracking lateralized hand use. The authors point to a captivating phylogenetic continuity between language lateralization in humans and brain asymmetries related to gestures in nonhuman primates.

---

## [Decision Letter]

**Decision letter after peer review:**

Thank you for submitting your article "Broca area homologue's asymmetry reflects gestural communication lateralisation in monkeys (Papio anubis)" for consideration by *eLife*. Your article has been reviewed by 2 peer reviewers, and the evaluation has been overseen by Ingrid Johnsrude as the Reviewing Editor and Timothy Behrens as the Senior Editor. The following individual involved in review of your submission has agreed to reveal their identity: Cedric Boeckx (Reviewer #1).

Essential revisions:

1) A closer look at the raw depth data (which the authors have openly shared) suggests that variability at the endpoints of the sulcus, thus including the most ventral portion, is larger across subjects compared to the middle portion. Plotting the standard deviations is illustrative in this respect. The employed method does not exclude the possibility that this variability may result from difficulties in measuring the sulcus depth at the endpoints or due to alignments, for example when the sulci have different lengths. It is not clear how you ensured that the length of the sulcus in the left hemisphere overlaps with the one in the right hemisphere. Your method seems to assume that a certain portion of sulcus to the left (a bin) corresponds to the same portion of sulcus to the right. This assumption is fundamental to your analysis, right? In which case, is it possible that the significance observed in a portion of the sulcus is a "misalignment" artifact (the big standard deviations for this portion are concerning in that regard). Would it be possible to back up the work with VBM analysis (eg Backer et al. Brain Struct Funct 2021)? The VBM analysis might not be necessary if they are able to account for the concern in another way.

2) The correlations reported in Figure 2 might suffer from circularity problems, since the analysis is based on the AQ scores derived from the subset of bins which were already shown as significant in Figure 1. This needs to be addressed.

3) What is the evolutionary link between left lateralization for language in humans and gestural communication in the current study? The suggestion of phylogenetic continuity between language-related frontal lateralization and gestural communication seems premature in light of the contralateral effects the authors report. In this regard, while a functional shift from motor regions to the more inferior frontal regions for the gestural communication system may seem relevant, it remains to be elucidated how structural asymmetry-which has been shown to be in both the left and right hemispheres in baboons-can be turn into a fully left-lateralized language system in humans. Views arguing against the gesture-first hypothesis of language evolution should therefore also be considered (e.g., Kendon Psychon Bull Rev 2017).

4. On line 52 they state that research on gestural communication in primates has recently received increased attention. The word 'recently' is inappropriate, since a gestural origin for language has long been entertained. Rephrase this part to acknowledge this research tradition ('gestural protolanguage hypothesis').

5. On line 75-77, the authors list a couple of publications to support the idea that "the neural substrates of typical handedness measures and language brain organisation might be not related but rather independent from each other". There are more recent and comprehensive studies that should be cited in this context (the works of N.Tsourio-Mazoyer and C. Franks).

6. Pointers to the literature on sign language and handedness in the context of gestural communication and language lateralization (line 78 et seq.) would be a helpful addition.

7. The end of the Discussion section could be better articulated. Lines 352-354 should be expanded upon: "In an evolutionary perspective we propose therefore that the intentional communicative function of primate's communicative gesture might be a functional interface between actions and language prerequisites, nested inside the monkeys' Broca's homologue." Likewise, the notion of "embodied modality" on line 356 may not be fully transparent to readers.

8. references to complementary work seeking deep roots for 'language'-related brain circuitry (e.g.: PMID: 32313267) [lines 359-361] should be added.

9. Fix the ungrammatical sentence fragment page. 8 LL: 173-174

"Similarly, then apes, some communicative manual gestures…".

---

## [Author Response]

Essential revisions:1) A closer look at the raw depth data (which the authors have openly shared) suggests that variability at the endpoints of the sulcus, thus including the most ventral portion, is larger across subjects compared to the middle portion. Plotting the standard deviations is illustrative in this respect. The employed method does not exclude the possibility that this variability may result from difficulties in measuring the sulcus depth at the endpoints or due to alignments, for example when the sulci have different lengths. It is not clear how you ensured that the length of the sulcus in the left hemisphere overlaps with the one in the right hemisphere. Your method seems to assume that a certain portion of sulcus to the left (a bin) corresponds to the same portion of sulcus to the right. This assumption is fundamental to your analysis, right? In which case, is it possible that the significance observed in a portion of the sulcus is a "misalignment" artifact (the big standard deviations for this portion are concerning in that regard). Would it be possible to back up the work with VBM analysis (eg Backer et al. Brain Struct Funct 2021)? The VBM analysis might not be necessary if they are able to account for the concern in another way.

Thank you for this in-depth check of our data. In fact, "misalignment" artifact between left and right hemisphere are very unlikely to explain our effect for the 3 following reasons:

First, for each sulcus, the automated sulci parametrization algorithm in BrainVisa provides depth measurements for 99 equal cuts within the entire sulcus, normalizing thus the length of both the left and the right sulcus, which prevent thus any potential "misalignment" artifact. In other word, it means that the position 33 in the right sulcus and the position 33 in the left sulcus are corresponding exactly to the same 1/3 portion of each sulcus across all subjects, regardless of their respective length. So the significant normalized bin found at the last third section of the sulcus (i.e., from 65 to 95) is corresponding to the exact relative sulcus portion in the two L/R sulci and in the 50 subjects.

Second, the specific effect occurs for, not only in the extremity, but in large portion (about the last third portion) of the sulcus (i.e. from positions 65 to 95).

Finally, the effect occurs exclusively for handedness for communicative gesture but not for handedness for non-communicative manipulation. There is no reason why any "misalignment" artifact would have impacted only asymmetry sulcal effect for gestural communication and not similarly for handedness for manipulation. Actually, the AQ variability that the reviewer noticed at the most ventral portion, is the main focused of the study for which we provided solid evidence that such a AQ variability is related to variation of handedness direction across subjects for gestural communication exclusively.

A VBM study would be very interesting for replicating our results. Therefore, we added this VBM perspective in the discussion:

“Since the sulcus depth might reflect a gyral surface and its underlying grey matter volume, future work of delineating and quantifying grey matter of the ventral IA sulcus would help determining which of those sub-regions of the Broca homologue is driving the asymmetry specifically, for instance by VBM methods.”

Although interesting, we don’t see the need or relevance for conducting, in the present paper, an entire new VBM study – as a simple “back up” – which, by the way, has its own limitations (Mechelli et al., 2005), especially for brain asymmetry studies regarding interhemispheric misalignment issue. Sulci depth approach on brain asymmetry has been acknowledged as solid and mature enough for comparative research (e.g., Coulon et al., IEEE EMBS 2011; Cykowski et al., Cerebral Cortex 2008; Hopkins et al., J. of Neuroscience 2010; Leroy et al., PNAS 2015). We demonstrated our findings are robust and stand on its own, the work being carefully controlled by both behavioral (non-communicative task) and sulcus (central sulcus) controls.

2) The correlations reported in Figure 2 might suffer from circularity problems, since the analysis is based on the AQ scores derived from the subset of bins which were already shown as significant in Figure 1. This needs to be addressed.

It is true that the variables “HI scores” (degree of hand preference) used in the correlation analysis were used also to define the category of handedness (i.e., left- or right- direction of handedness) used in the first categorial analysis. But, the respective finding provided by categorial analysis versus by correlational analysis are both valuable since different and complementary, given direction and degree effects might be independent and inconsistent. In other words, finding an effect of the direction of handedness (left versus right-hand) on the AQ scores in the categorial analysis doesn’t not mean that the degree of manual asymmetry is correlated to the degree of brain asymmetry. So additional correlational analyses are needed to test it (see Margiotoudi et al., 2019 Cortex). And vice et versa: the finding – that the stronger the degree of hand preference (HI), the stronger the degree of brain asymmetry (AQ) – is a valuable additional information that does not imply that the left-right direction of handedness has an influence on the AQ values. In fact, we found both, reinforcing the asymmetric link we found between gestural communication behaviour and the ventral third part of the inferior arcuate sulcus at both a qualitative and quantitative levels. We added in the discussion that this correlation analyses are built upon the first categorial findings:

“Building upon these first results, we also found a significant negative correlation between the Handedness Index (HI) values for gestures and the Asymmetric Quotient (AQ) depth values of the IA sulcus “Broca cluster”, suggesting that the contralateral links between handedness for gestural communication and depth asymmetries at the most ventral portion of the IA sulcus is evident not only at a qualitative level but also at a quantitative level as well.”

3) What is the evolutionary link between left lateralization for language in humans and gestural communication in the current study? The suggestion of phylogenetic continuity between language-related frontal lateralization and gestural communication seems premature in light of the contralateral effects the authors report. In this regard, while a functional shift from motor regions to the more inferior frontal regions for the gestural communication system may seem relevant, it remains to be elucidated how structural asymmetry-which has been shown to be in both the left and right hemispheres in baboons-can be turn into a fully left-lateralized language system in humans. Views arguing against the gesture-first hypothesis of language evolution should therefore also be considered (e.g., Kendon Psychon Bull Rev 2017).

Thank you for this point which indeed requires clarification in our manuscript. We added thus a paragraph at the end of the introduction to clarify the implications of our contribution in comparison to previous comparative population-level works on leftward brain lateralization for language.

“In sum, within the framework of the origin of hemispheric specialisation for language, most comparative works in nonhuman primates focused on population-level asymmetry for either brain or communicative behaviours. Those studies have reported similar population-level leftward brain asymmetry for key language homologue regions (Gannon et al., 1998; Catalupo and Hopkins, 2001; Marie et al., 2018; Becker et al., 2021 b,c) or similar populational-level right-handedness for communicative gestures (reviewed in: Meguerditchain et al., 2013). Nevertheless, to test potential phylogenetic continuities, this approach suffered from lack of studies investigating the direct behavioural/brain correlates at the individual-levels.”

It must be noted that not all humans are functionally left lateralized for language neither and structural left lateralisation of key-language regions like the Planum Temporale occur in about 65% of the (human) population just like in nonhuman primates (e.g., Geschwind and Levitsky, Science 1968).

Difference in degree in the population-level leftward lateralization between baboons and humans does not mean that there is no continuity in the underlying behavioral/brain correlates at the individual level. It is probable that a population-level bias, which is present in the baboon, has increased in humans in the course of the evolution.

Regarding views arguing for multimodal language origins (like Kendon, 2017), we have added in our discussion:

“…suggests that gestural communication may be a compelling modality for one of the multimodal evolutionary roots of the typical multimodal language system in humans and its left-hemispheric specialization.”

4. On line 52 they state that research on gestural communication in primates has recently received increased attention. The word 'recently' is inappropriate, since a gestural origin for language has long been entertained. Rephrase this part to acknowledge this research tradition ('gestural protolanguage hypothesis').

Ok, good point, we deleted the word “recently” and reformulate.

We have rephrased accordingly:

“the research on gestural communication in apes and monkeys has historically shown a significant interest within this evolutionary framework.”

5. On line 75-77, the authors list a couple of publications to support the idea that "the neural substrates of typical handedness measures and language brain organisation might be not related but rather independent from each other". There are more recent and comprehensive studies that should be cited in this context (the works of N.Tsourio-Mazoyer and C. Franks).

We have added line 73: Labache et al., *eLife* 2020 to the list of publications.

6. Pointers to the literature on sign language and handedness in the context of gestural communication and language lateralization (line 78 et seq.) would be a helpful addition.

We have added in line 80:

“Moreover, further work showed Broca activation in the left hemisphere also for sign language production including manual and oro-facial gestures (Emmorey et al., 2004; MacSweeney, and Waters, 2008).”

7. The end of the Discussion section could be better articulated. Lines 352-354 should be expanded upon: "In an evolutionary perspective we propose therefore that the intentional communicative function of primate's communicative gesture might be a functional interface between actions and language prerequisites, nested inside the monkeys' Broca's homologue." Likewise, the notion of "embodied modality" on line 356 may not be fully transparent to readers.

We have rephrased the last paragraph of our discussion accordingly:

“Regarding Broca’s area in humans, very recently, a functional segregation was proposed with Broca’s anterior part implicated in language syntax and its posterior part exclusively implicated in motor actions (Zaccarella et al., 2021). The authors argued that action and language meet at this interface. In an evolutionary perspective we propose therefore that the intentionality of primate’s communicative gesture might account for this hypothesized functional interface of actions and language prerequisites, nested inside the monkeys’ Broca’s homologue (see also: Arbib 2006, Rizzolatti 2017, Corballis, 2015). In addition, in macaques Broca’s homologue, neuronal recordings showed populations of specific neurons activated for both volitional vocal and manual actions (Gavrilov and Nieder, 2021). The articulation of our results with this recent literature suggests that gestural communication may be a compelling modality for one of the multimodal evolutionary roots of the typical multimodal language system in humans and its hemispheric specialisation. It is thus not excluded that language-related frontal lateralisation might be much older than previously thought and inherited from a gestural communicative system dating back, not to Hominid origins, but rather to the common ancestor of humans, great apes and Old World monkeys, 25–35 million years ago.”

8. References to complementary work seeking deep roots for 'language'-related brain circuitry (e.g.: PMID: 32313267) [lines 359-361] should be added.

We have added this reference:

“(…) whereas comparative language research has focused on the potential continuities across primate brain circuitry (e.g., Balezeau et al., 2020; Becker et al., 2021a) or vocal and auditory systems (…)”

9. Fix the ungrammatical sentence fragment page. 8 LL: 173-174"Similarly, then apes, some communicative manual gestures…".

Thanks, we have changed the sentence:

“Like in apes, some communicative manual gestures in baboons have been found to share human-like intentional control, referential properties, flexibility of acquisition and of use as well as similar specific pattern of manual lateralisation (reviewed in Tomasello, 2008; Meguerditchian and Vauclair, 2014; Meguerditchain et al., 2013).”